# Analysis of Wind Pressure Coefficients for Single-Span Arched Plastic Greenhouses Located in a Valley Region Using CFD

Zongmin Liang [1], Guifeng He [1], Yanfeng Li [2], Zixuan Gao [1], Xiaoying Ren [1], Qinan Wu [3], Shumei Zhao [1] and Jing Xu [1],*

1   College of Water Resources and Civil Engineering, China Agricultural University, Beijing 100083, China
2   Vegetable Research Institute of College of Agriculture and Animal Husbandry of Tibet Autonomous Region, Lhasa 851418, China
3   Tibet Academy of Agricultural and Animal Husbandry Sciences, Lhasa 851418, China
*   Correspondence: xujing@cau.edu.cn

**Abstract:** The wind pressure coefficient is essential for calculating the wind loads on greenhouses. The wind pressure on single-span arched greenhouses built in valleys differs from those in plain regions. To promote our understanding of wind characteristics and ensure the structural safety of greenhouses in valley areas, an analysis of the distribution law of wind pressure on greenhouses is required. Firstly, we carried out a survey on greenhouse distribution and undulate terrain distribution near greenhouses in Tibet and measured the air density in Lhasa, Tibet. Then, employing the validated realizable $k$-$\varepsilon$ turbulence model and the verification of grid independence, the wind pressure on greenhouses with different greenhouse azimuths was investigated. According to the survey results, values, such as the distance between the greenhouse and the mountain in addition to the greenhouse azimuth, were also obtained for calculating the wind pressure on greenhouses placed in valleys. A calculation model considering the relationship between the mountain distance and the wind pressure coefficient is proposed, whose results fit well with the results from computational fluid dynamics. The relative errors between the two different results are within 15%. Research shows that there is a canyon wind effect in the valley area, and its effect on wind pressure should be considered in greenhouse design. This research is valuable for the design of plastic greenhouses built in Tibet or other valley regions.

**Keywords:** valley topography; plastic greenhouse; wind pressure coefficient; computational fluid dynamics; calculation model





## 1. Introduction

In recent years, the government of China has encouraged the development of plastic greenhouses and solar greenhouses in valley terrain. Such greenhouses should be precisely designed because they are easily destroyed by strong winds [1]. Currently, the design standard used for wind load in a valley is mainly based on design experience or the specifications for inland plains in China. However, canyon wind exists in valley areas [2–4], which is quite different from that in plain areas [5–7]. It is, therefore, not reasonable to design greenhouses in valley areas according to the load codes in plain areas of China. Therefore, the effects of valley wind load on greenhouses should be carefully analyzed to secure the safety of greenhouses and to provide data as a basis for the revision of standards.

Wind load is closely related to the wind pressure coefficient $C_p$ [8,9], whose formula was presented in Xu et al. [10]. There are currently three main methods to determine wind pressure coefficients, i.e., field experiment, wind tunnel test, and computational fluid dynamics (CFD). These methods were compared and presented, and the effects of greenhouse type, greenhouse roof angle and roof spacing were discussed respectively to determine the wind pressure on low-rise buildings or greenhouses in [11–14]. In these documents,

only Wang et al. [14] mentioned the impact of topographic factors such as mountain terrain on greenhouse partial pressure, but they still did not give a specific research plan. Wind pressure on different buildings has been studied through field experiments [15,16], but this method is not broadly applicable due to the unpredictability and instability of the external climate environment.

Wind tunnel experiments are commonly conducted to determine the wind pressure on greenhouses, and they can be used to measure the effects of different parameters on wind pressure [17–19]. They discussed the influence of the number, arrangement, support and size of greenhouses on the wind pressure and gave suggestions on the decision of the wind coefficient. However, the test settings of environmental conditions in the last two articles have certain limitations: they are only applicable to the relatively open environment. Wind tunnel tests [20–23] are also applied to investigate the effect of ridge height or wind angle on the distributions of the wind pressure acting on greenhouses. Bautista et al. [21] compared the wind tunnel test results with CFD calculation results, but the wind direction angle is single during the test, and the test environment is too ideal. The wind pressure coefficients of four typical single-span greenhouses used in Korea were measured in the wind tunnel according to wind direction [22], which can be used as a reference to calculate and test the wind pressure coefficient on greenhouses in valleys. Bronkhorst et al. [23] studied the characteristics of the wind pressure coefficient on multi-span greenhouses and used the average wind pressure coefficient in the calculation, resulting in a conservative and non-relevant result. Meanwhile, wind tunnel tests also have the limitations of restrained greenhouse model size due to the blockage ratio, high expense of making models, and large consumption of time and labor [1].

Due to these disadvantages, computational fluid dynamics (CFD) studies have been developed to evaluate the wind pressure on greenhouses or low-rise structures [1,24–26]. CFD was used to gain the wind pressure coefficients on greenhouses with different span-height ratios [27]. Wind pressure values with various spans of Venlo-type greenhouses have also been analyzed based on turbulence models [28]. To affirm the accuracy and precision of the numerical simulation model, many study results on CFD model evaluation [8,10,29] have been compared with the results of the field experiments of Wells and Hoxey [30] and Hoxey and Richardson [31,32] or wind tunnel measurements of Robertson et al. [33]. It is not difficult to see that scholars have been very skilled in using CFD to study wind pressure, but there are still many problems in some aspects. For example, compared with the wind pressure characteristics of the greenhouse cover surface, more research focuses on the gas flow inside the greenhouse and the operating conditions of agricultural buildings [24,25]; When Neto et al. [27] used CFD model to predict the wind pressure coefficient of greenhouses with different height-span ratio, he only considered the specific valley wind direction, which made certain limitations in helping greenhouse construction; In addition, some results obtained from field tests and wind tunnel tests have lost credibility due to long test time, errors caused by manual operation of measuring instruments and other factors [28,29].

Nevertheless, the above-mentioned studies were conducted in plain areas. Many scholars [34–37] have performed different methods to study the characteristics of valley wind flow. Catalano and Moeng [34] used WRF patterns and a subgrid-scale turbulence scheme to discuss the characteristics of valley flow. However, the research did not involve a wind pressure study. Serafin and Zardi [35] gave the influence of slope wind and turbulence on the surface and atmospheric core of the ideal valley in the daytime, but their research focused on heat transfer, not on wind pressure, whose result cannot be fully referenced. Cao et al. [36] investigated the turbulent boundary layer over two-dimensional hills by performing a large-eddy simulation (LES). However, in this study, the slope of the selected hill model is relatively flat, and the research results are not completely applicable to the region with a large undulation of the hill slope. Takeo et al. [38] studied a small-scale model on a solid hill based on a wind tunnel test, but due to the small scale, the measurement result is not universal. Martins et al. [39] measured the wind characteristics of complex

terrain in Southern Brazil according to a field test. These researchers, however, were all concerned about wind flow in valley areas. Less research has been conducted on how the wind pressure coefficient on a greenhouse is affected by valley wind [40].

In our study, the mountain model and structural model are placed in the same wind flow field. The aim is to evaluate the wind pressure coefficient of a single-span arched plastic greenhouse built in a valley area. In the study, different turbulence models are verified based on the field experiment of Richards and Hoxey [15] and the wind tunnel test of Murakami and Mochida [41]. Realizable $k$-$\varepsilon$ model is finally chosen for the simulation. In addition, a grid independence test is conducted to ensure the correctness of the model. Based on the selected turbulence model and grid independence test, CFD model evaluations are carried out by the usage of a wind tunnel test [22] to improve the reliability of the CFD-designed model. Different parameters, such as the greenhouse azimuth and the distance between the hill and the greenhouse, are considered in the simulation. Calculation models are then proposed by considering the above parameters. The research results can be applied to the structural design of plastic greenhouses and serve as a supplement for the revision of load specifications in China [42].

## 2. Model Dimensions and Model Establishment

### 2.1. Mountain Model Parameters

To determine the dimension of the mountain and structural models for the valley area, the construction of plastic greenhouses was investigated in Tibet Autonomous Region, which is an alpine and high-altitude region with special geographical conditions. The investigation on the construction situation of greenhouse clusters shows that there is mainly a total of 67 larger greenhouse clusters in Tibet, including 10 clusters in Shigatse (No. 1–10), 21 clusters in Lhasa (No. 11–31), and 36 clusters in Nyingchi (No. 32–67).

Table 1 shows the number of greenhouse clusters considering the distances between the outermost greenhouse and the mountain, which means that there are variations in the distances between greenhouses and mountains. Thus, for the simulation, distances of 0–900 m were chosen. Table 2 shows the number of clusters with different greenhouse azimuths. Thus, greenhouse azimuths of 45° and 90° were selected in this study. Research on an azimuth of 0°has been presented by Xu et al. [10].

**Table 1.** The number of greenhouse clusters according to different distances from the mountain.

| Distance between the Greenhouse and the Mountain (m) | 0–50 | 50–100 | 100–200 | 200–300 | 500–1000 | 1000–2000 |
|---|---|---|---|---|---|---|
| Number of greenhouse clusters | 20 | 6 | 11 | 15 | 10 | 5 |

**Table 2.** The number of greenhouse clusters according to different greenhouse azimuths.

| Angle between Greenhouse and Valley $\theta$ (°) | $\theta = 0$ | $0 < \theta < 90$ | $\theta = 90$ |
|---|---|---|---|
| Number of greenhouse clusters | 8 | 34 | 25 |

Table 3 presents the number of clusters with different mountain slopes, which shows that in most cases, the mountain slope is between 30 and 45 degrees. Due to the complicacy of building a real valley terrain in CFD, the mountain is simplified as a sinusoidal geometrical shape $z = H[1 + \cos(\pi r/2L_1)/2]$ with a slope angle of 45 degrees to obtain the characteristics of wind load on greenhouses in dales. In the equation $r = \sqrt{x^2 + y^2}$, $H$ is mountain height, and $L_1$ is the horizontal distance from the mountainside to the summit (Figure 1). In the modulation, the height $H$ equals 100 m, the distance $L_1$ is equal to 100 m, the diameter $D$ of the hill is equal to 400 m, and the length of the ridge is 300 m (Figure 2a). The slope of the mountain is 0.5.

**Table 3.** The number of greenhouse clusters according to different mountain slopes.

| Mountain Slope (°) | 0–15 | 15–30 | 30–45 | 45–60 |
|---|---|---|---|---|
| Number of greenhouse clusters | 8 | 20 | 27 | 12 |

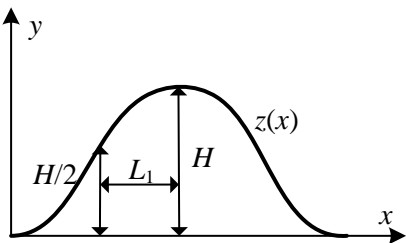

**Figure 1.** Parameter diagram of contour line equation of the sinusoidal model.

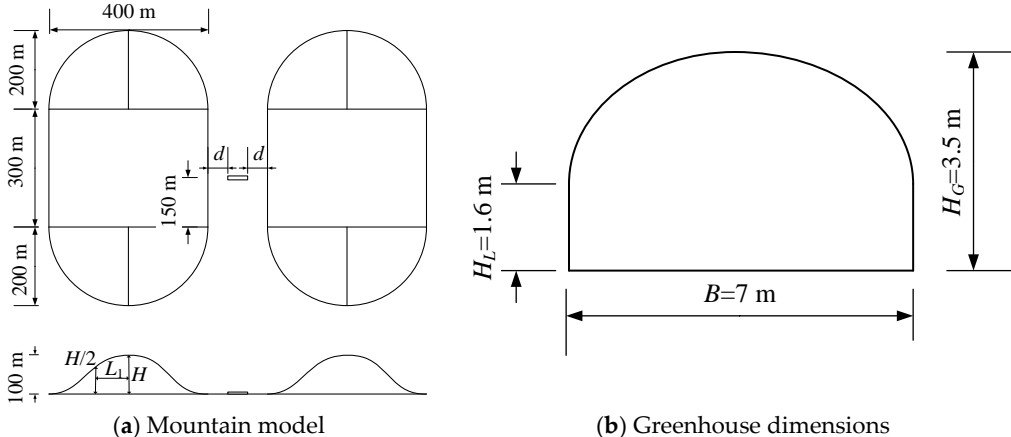

(**a**) Mountain model      (**b**) Greenhouse dimensions

**Figure 2.** Size and layout of greenhouse and mountain.

### 2.2. Dimension of Structure Model

The greenhouse was placed in the middle of 2 mountains. We defined *d* as the horizontal distance between the foot of the hill and the greenhouse (Figure 2a). A single-span arched plastic greenhouse was chosen from the most extensive dimensional models in China. The length *L*, height $H_G$, and span of the greenhouse *B* were 44 m, 3.5 m, and 7 m, respectively. The canopy shoulder height $H_L$ was 1.6 m (Figure 2b).

## 3. Model Verification

### 3.1. Turbulence Model Validation

To determine the turbulence model used in the simulation, six common turbulence models, including the standard *k-ε*, realizable *k-ε*, RNG *k-ε*, standard *k-ω*, SST *k-ω*, and BSL models, were selected. Through comparative analysis of the results of the tunnel tests [41] and field experiments [15], six turbulence models were compared and validated to evaluate the quality of models and to provide a basic model for the subsequent simulation analysis. In the simulation, the geometric dimensions of the model, the entrance wind profile, and turbulence intensity are exactly the same as in the wind tunnel test from Murakami [41]; the inlet boundary conditions are the same as in Wang [43]. The calculation flow field is $21H \times 11H \times 6H$, where *H* is the height of the greenhouse, and the requirement of a blocking rate of less than 3% is met. The total number of meshing grid and summarize points are 1,045,183 and 1,080,160, respectively. The convergence residuals are set to $1 \times 10^{-4}$ as the convergence criteria, and the changes in wind pressure are monitored until it is stable.

Figure 3 shows the wind pressure on the surface of the structure model using six different turbulence models. It is shown that the results of the six kinds of turbulence

models in the simulation of the wind pressure distribution rule in the center of the cube are roughly the same as those from the wind tunnel test and field test. A comparison of the field test results, wind tunnel results, and simulation results from the six turbulence models is shown in Table 4.

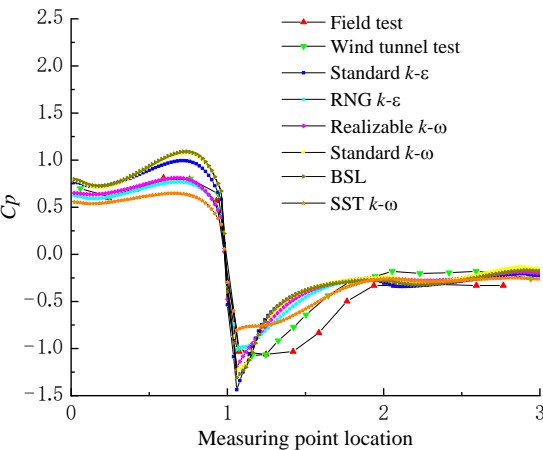

**Figure 3.** Wind pressure distribution curves with different turbulence models.

**Table 4.** Wind pressure coefficients from different methods.

| Abscissa Value | Field Test | Wind Tunnel Test | Standard $k$-$\varepsilon$ | RNG $k$-$\varepsilon$ | Realizable $k$-$\varepsilon$ | Standard $k$-$\omega$ | BSL | SST $k$-$\omega$ |
|---|---|---|---|---|---|---|---|---|
| 0.40 | 0.690 | 0.698 | 0.823 | 0.653 | 0.702 | 0.836 | 0.845 | 0.585 |
| 0.60 | 0.812 | 0.793 | 0.958 | 0.705 | 0.798 | 1.019 | 1.027 | 0.533 |
| 0.94 | 0.568 | 0.643 | 0.660 | 0.452 | 0.467 | 0.749 | 0.752 | 0.350 |
| 1.08 | −1.031 | −0.994 | −1.340 | −0.790 | −1.137 | −1.204 | −1.271 | −0.798 |
| 1.24 | −1.061 | −1.065 | −0.695 | −0.763 | −0.771 | −0.732 | −0.715 | −0.753 |
| 1.42 | −1.031 | −0.775 | −0.440 | −0.552 | −0.504 | −0.445 | −0.440 | −0.640 |
| 1.94 | −0.331 | −0.235 | −0.274 | −0.242 | −0.250 | −0.249 | −0.273 | −0.271 |
| 2.06 | −0.331 | −0.180 | −0.335 | −0.268 | −0.260 | −0.251 | −0.292 | −0.259 |
| 2.26 | −0.315 | −0.203 | −0.329 | −0.249 | −0.279 | −0.300 | −0.321 | −0.285 |
| 2.60 | −0.331 | −0.180 | −0.260 | −0.204 | −0.237 | −0.219 | −0.245 | −0.266 |
| 2.76 | −0.331 | −0.211 | −0.225 | −0.137 | −0.203 | −0.161 | −0.195 | −0.252 |
| Errors with the field test | | | 0.242 | 0.207 | 0.196 | 0.240 | 0.245 | 0.204 |
| Errors with the wind tunnel test | | | 0.204 | 0.149 | 0.144 | 0.182 | 0.198 | 0.177 |

As illustrated in Table 4, the results derived from realizable $k$-$\varepsilon$ were more identical to those from the wind tunnel and field tests. The error between the realizable $k$-$\varepsilon$ and the field test is 0.196, and the error between the realizable $k$-$\varepsilon$ and the wind tunnel test is 0.144, which is the lowest of such values for all six models; thus, realizable $k$-$\varepsilon$ was chosen for the following simulation. In the comparison, the equation of error is

$$\text{error} = \sqrt{\frac{1}{n}\sum_{n}\left(C_{p,\text{Test}} - C_{p,\text{Turbulence-model}}\right)^2} \tag{1}$$

where $n$ is the sample number, $C_{p,\text{Test}}$ is the wind pressure coefficient obtained from the wind tunnel experiment and the field test, and $C_{p,\text{Turbulence-model}}$ is the wind pressure coefficient obtained from six turbulence models.

In addition, when the time average strain rate is large, negative pressure may appear in the simulation using the standard $k$-$\varepsilon$ model. Therefore, some mathematical constraint on the positive pressure is needed to ensure the flow conforms to the physical laws of turbulence. Based on the above principles, a realizable $k$-$\varepsilon$ model was proposed, and the

relevant formula was presented by Shih et al. [44]. This model mainly introduces the contents related to rotation and curvature. In the formula of this model, the penultimate term in the equation for $\varepsilon$ has no singularity, even if the value of $k$ is small or equal to zero, and the denominator will not be zero. Therefore, the model shows good performance in rotational flows, boundary layer flows with strong adverse pressure gradients, flow separation, and secondary flows [45]. Above all, considering the separation and rotation flow generated by wind passing through the mountain and the greenhouse, the realizable $k$-$\varepsilon$ turbulence model was selected for analysis.

### 3.2. Fluid Domain and Grid Validation

The arrangement of the single-span arched greenhouse and the mountain is shown in Figure 4. The dimensions of the flow field are $6\,H \times 7\,H \times 15\,H$. The model is located at the front 1/3 of the computational domain of the wind flow field. To reduce the interaction between fluid and boundary, the blocking rate of the model is set to less than 3% [8].

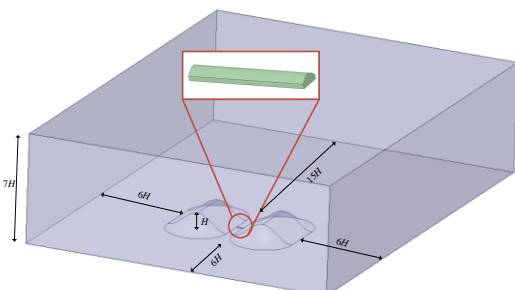

**Figure 4.** Layout of the model and size of the fluid domain size.

To enhance the accuracy and efficiency of the simulation results, the grids around the mountains and greenhouse are regionally densified. Five grid independence tests are conducted to determine the mesh size. The greenhouse and the mountain are divided into different surface grid sizes M1, M2, M3, M4, and M5. The size of each is $l/10$, $l/15$, $l/20$, $l/25$, and $l/30$, respectively, where $l$ is the minimum characteristic size of the model (for the greenhouse, $l = H_L = 1.6$ m, for the mountain, $l = H_1 = 100$ m). The grid sizes are shown in Table 5. The growth rate of the surface grid is set as 1.05, and the boundary layer as 6 layers, as shown in Figure 5. To better capture the flow of the wind field around the greenhouse and mountain, the volume mesh growth rate is set to 1.1.

**Table 5.** Number and dimensions of the grids.

| Size of Mesh | M1 | M2 | M3 | M4 | M5 |
|---|---|---|---|---|---|
| Meshing size of greenhouse (m) | 0.160 | 0.106 | 0.080 | 0.064 | 0.053 |
| Meshing size of mountain (m) | 10.000 | 6.667 | 5.000 | 4.000 | 3.333 |
| Total number of grids | 639,114 | 1,258,233 | 1,917,194 | 2,860,792 | 4,028,784 |

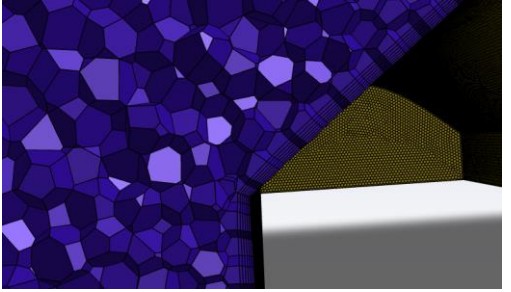 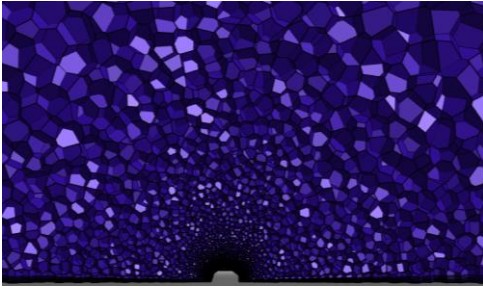

(**a**) Wall boundary layer mesh          (**b**) Fluid domain volume mesh

**Figure 5.** Schematic diagram of polyhedron mesh generation and boundary layer mesh.

To obtain the wind pressure on a single-span arched greenhouse with different grid sizes, the wind pressure coefficients at the middle of the greenhouse with 0° greenhouse azimuth angle are shown in Figure 6. On the windward side 0−1, the numerical simulation results of the M1 grid are quite different from those of the other four grids because the element size of the M1 grid is relatively rough. For locations 1–2 and 2–3, the simulation results of the five grids show good consistency. Therefore, considering both the computing efficiency and time cost, the M2 grid is selected in the following simulation, whose dimensions of the meshing grid for the greenhouse and mountain are 0.106 and 6.667 m, respectively. The largest grid is 50 m, and the number of grids ranges from 1.20 million to 2 million.

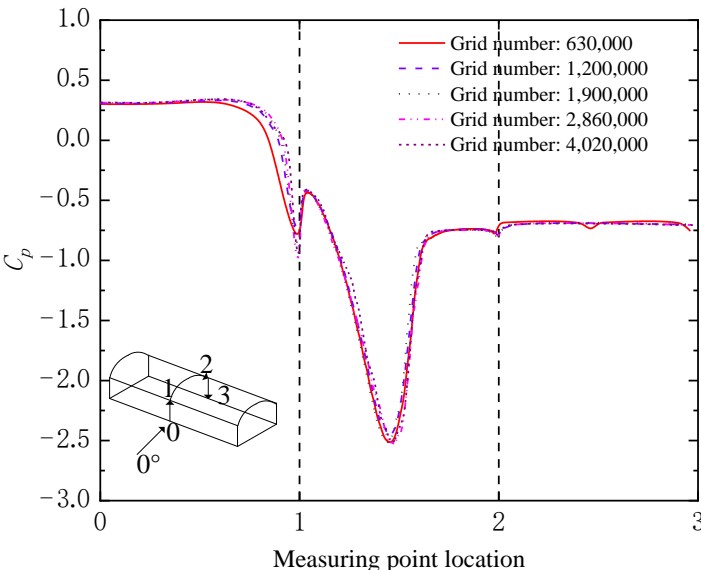

**Figure 6.** Comparison of wind pressure coefficients for different grid sizes.

### 3.3. Boundary Conditions of the Flow Field

The velocity inlet is defined as the inflow boundary condition. Because the parameters, such as exponential wind function, turbulent kinetic energy, and turbulent dissipation rate, cannot be directly set into the simulation model, the user-defined functions (UDF) are programmed.

As the turbulence intensity is not clearly defined in Chinese codes, the turbulence kinetic energy $k$ and turbulence dissipation rate $\varepsilon$ are gained from a Japanese specification [46]. The formulas for exponential wind speed $V$, turbulence kinetic energy $k$, and turbulence dissipation rate $\varepsilon$ are from Xu et al. [10].

Pressure-outlet is set as the outlet condition in the flow field, in which the normal gradient along the outlet direction for each physical quantity is zero. On the sides and top of the fluid domain, symmetrical boundaries are employed.

The realizable $k$-$\varepsilon$ model adopted in this paper is a high Reynolds number turbulence model and should be used together with a wall function. Accurate near-wall [47] modeling is important for successful CFD simulation because the solution gradients are notably high in wall-bounded flow. For the wall function, it is a general requirement that the mesh size of the first layer meets the condition 30 < y+ < 300 [48]. In order to ensure the accuracy of the solution, for this study, we chose scalable wall function (SWF), which can effectively improve the computational instability caused by the swaying of the first layer grid nodes between the viscous sublayer and the core layer during computational iteration processes. Non-sliding scalable wall functions are utilized at the bottom of the fluid domain. Figure 7 shows the y+ value of the greenhouse surface with 45°- and 90°-shed azimuth. It can be seen that although the distribution of y+ values on the surface of the greenhouse is irregular, it generally meets the requirements of 30 < y+ < 300. It shows that the mesh height in

the first layer on the wall meets the requirements of Scalable Wall Functions, ensuring the rationality of the calculation results. The algorithm adopts the default coupled algorithm in Fluent. In iterative computation, the iteration ends when the error is less than $1 \times 10^{-5}$, which means the flow field enters a steady state.

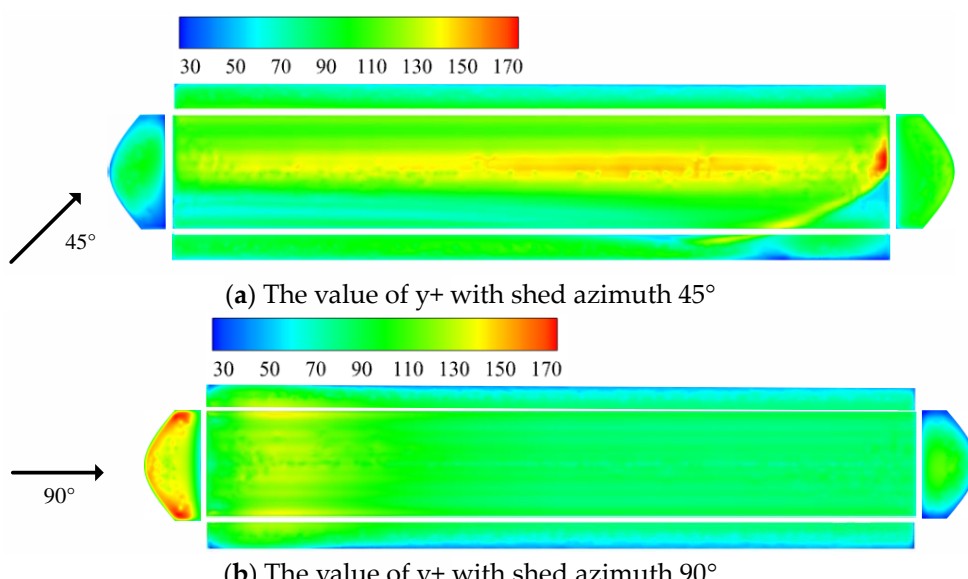

(**a**) The value of y+ with shed azimuth 45°

(**b**) The value of y+ with shed azimuth 90°

**Figure 7.** The value of y+ on the surface of the greenhouse.

### 3.4. Model Verification

Based on Section 3.1, the realizable *k-ε* turbulence model is selected for simulating the atmospheric boundary layer (ABL). To demonstrate the rationality of the realizable *k-ε* model and the selected meshing method, the test model of Kwon et al. [22] is simulated in CFD, and the wind pressure distribution on a greenhouse with either 0° or 90° wind angle derived from our study are displayed (Figures 8 and 9).

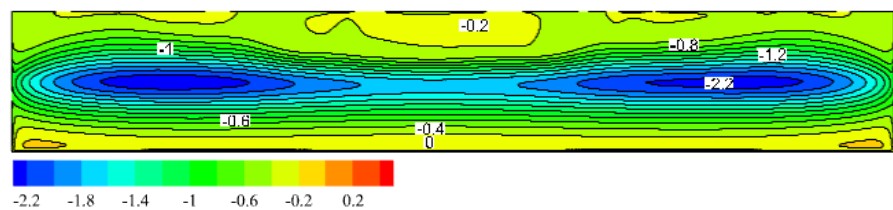

**Figure 8.** Wind pressure with 0°-wind angle derived from this paper.

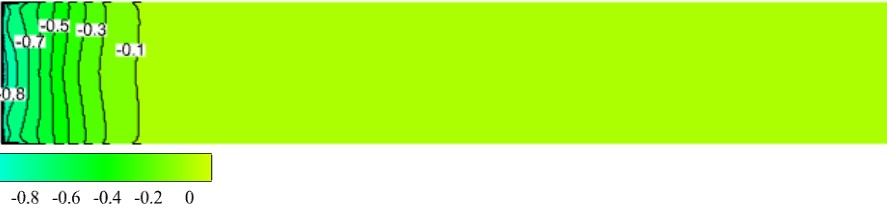

**Figure 9.** Wind pressure with 90°-wind angle derived from this paper.

According to the comparison of Figures 8 and 9 with Kwon et al. [22], the general regulation of the wind pressure distribution in greenhouses from the simulation model is in accordance with the results of Kwon et al. [22]. The greatest positive/negative pressure values occur in the same position. However, there are more details for the distribution of wind pressure obtained from the simulation than that from the wind tunnel test. In

summary, the selected realizable *k-ε* model and the proposed meshing method in this study are more appropriate than the wind tunnel test.

### 3.5. Model Partition

Utilizing the selected realizable *k-ε* model and meshing method, the distribution of wind pressure on a single-span arched greenhouse positioned in plain areas is illustrated in Figure 10. In combination with the distribution of wind pressure in Figure 10 and the standard for wind loads on roof structures [49], each surface of the arched shed is divided into zones (Figure 11). In Figure 11, the windward surface, leeward surface, roof surface, and sidewall are denoted by F, B, T, and W, respectively; the left side, middle side, and right side of the greenhouse are denoted by L, M, and R, respectively.

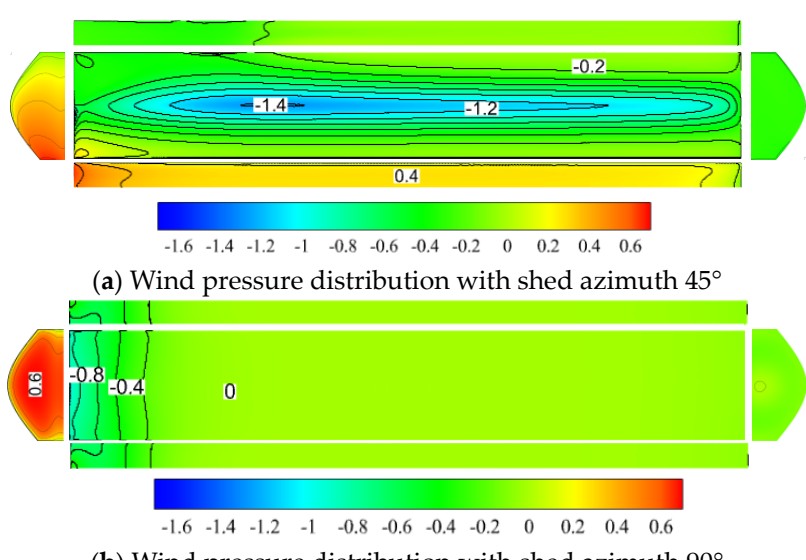

(**a**) Wind pressure distribution with shed azimuth 45°

(**b**) Wind pressure distribution with shed azimuth 90°

**Figure 10.** Distribution of wind pressure on single-span arched shed in a plain area.

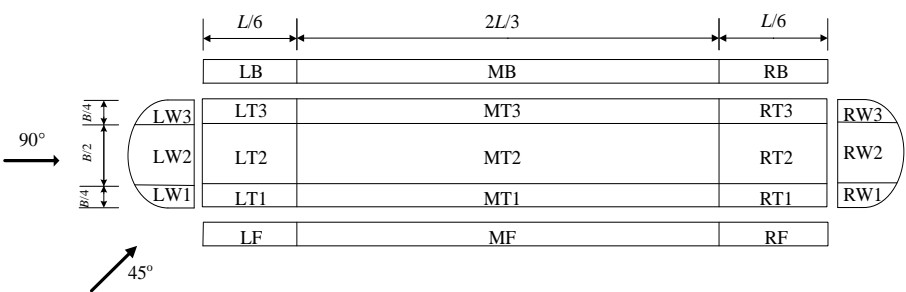

**Figure 11.** Zoning diagram of wind pressure coefficient with 45° and 90° azimuth angles.

## 4. Results and Discussions

### 4.1. Influence on Wind Pressure Coefficient of Shed Body

The flow diagrams of the wind field around the greenhouse and around the mountain are shown in Figure 12. An analysis of Figure 12 shows that when the wind passes through the mountain, part of the airflow is divided by the mountain and then passes back around the two sides of the mountain. When the separated airflow accumulates in the canyon area, a canyon wind effect is formed, which leads to increased wind speed in the canyon area. As a result, the influence of wind pressure on the greenhouse surface will be different from that on the plain. It can also be seen from Figure 12 that the maximum wind speed occurs at the top of the mountain, while the minimum wind speed occurs at the end wake of the valleys on both sides.

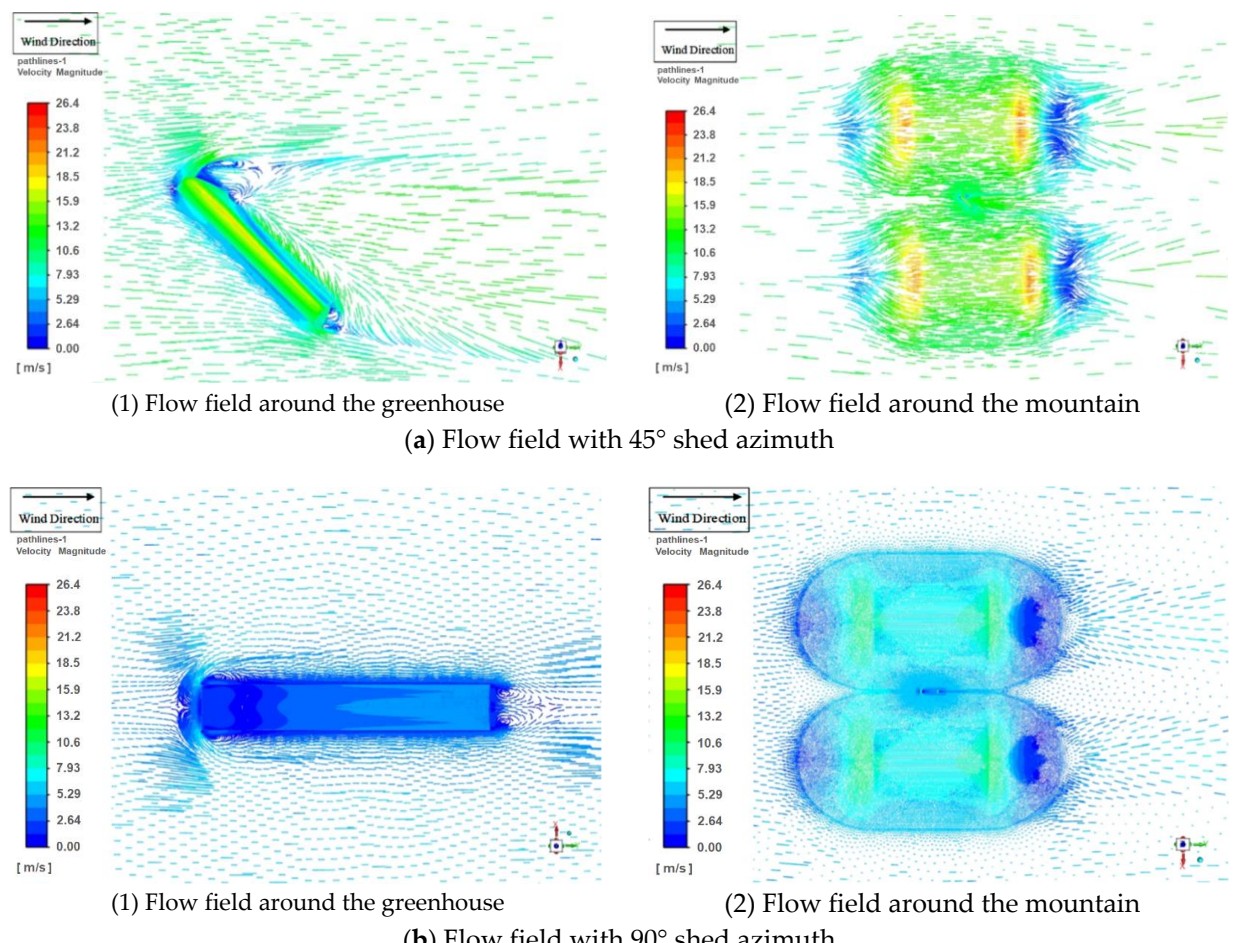

(1) Flow field around the greenhouse  (2) Flow field around the mountain

(**a**) Flow field with 45° shed azimuth

(1) Flow field around the greenhouse  (2) Flow field around the mountain

(**b**) Flow field with 90° shed azimuth

**Figure 12.** Flow field distribution on the surface of the shed.

When the greenhouse azimuth angle is 45°, the greenhouse forms a blocking effect on airflow. The airflow is separated at the intersection of the crosswind side and the windward side of the greenhouse. Part of the airflow bypasses the greenhouse roof and flows backward along the length of the greenhouse direction. It should be noted that from Figure 12a, when the azimuth angle is 45°, the airflow will produce a vortex in the MT2 area of the roof, resulting in negative pressure in this area. At the same time, chaotic vortices are produced in the LB area, and more vortices are also formed in the MB/RB area. When the azimuth angle of the greenhouse is 90°, the airflow will also separate on the windward side of the greenhouse. Part of the airflow will cross the roof and form a vortex on the windward side. Another part of the air flows to the back of the greenhouse. The two airflows converge at the end of the greenhouse and form two vortices on the lee side of the greenhouse.

Figure 13 shows the distribution of wind pressure on the greenhouse arranged in the valley and plain area. In the valley area, the horizontal distance $d$ between the greenhouse and the bottom of the mountains is equal to 0, 50, 150, 250, 350, 500, 700, and 900 m. According to Figure 12, the wind pressure coefficient shown in Figure 13 is weighted by Equation (2) to obtain the wind pressure coefficient on each zoning diagram of the greenhouse built in the valley region (see Figures 14 and 15) and the plain region (see Tables 6 and 7). The relationship between the ratios of wind pressure coefficient in the valley topography to that in plain area and the ratio of distance to the height of the mountain with different shed azimuths are shown in Figures 16 and 17.

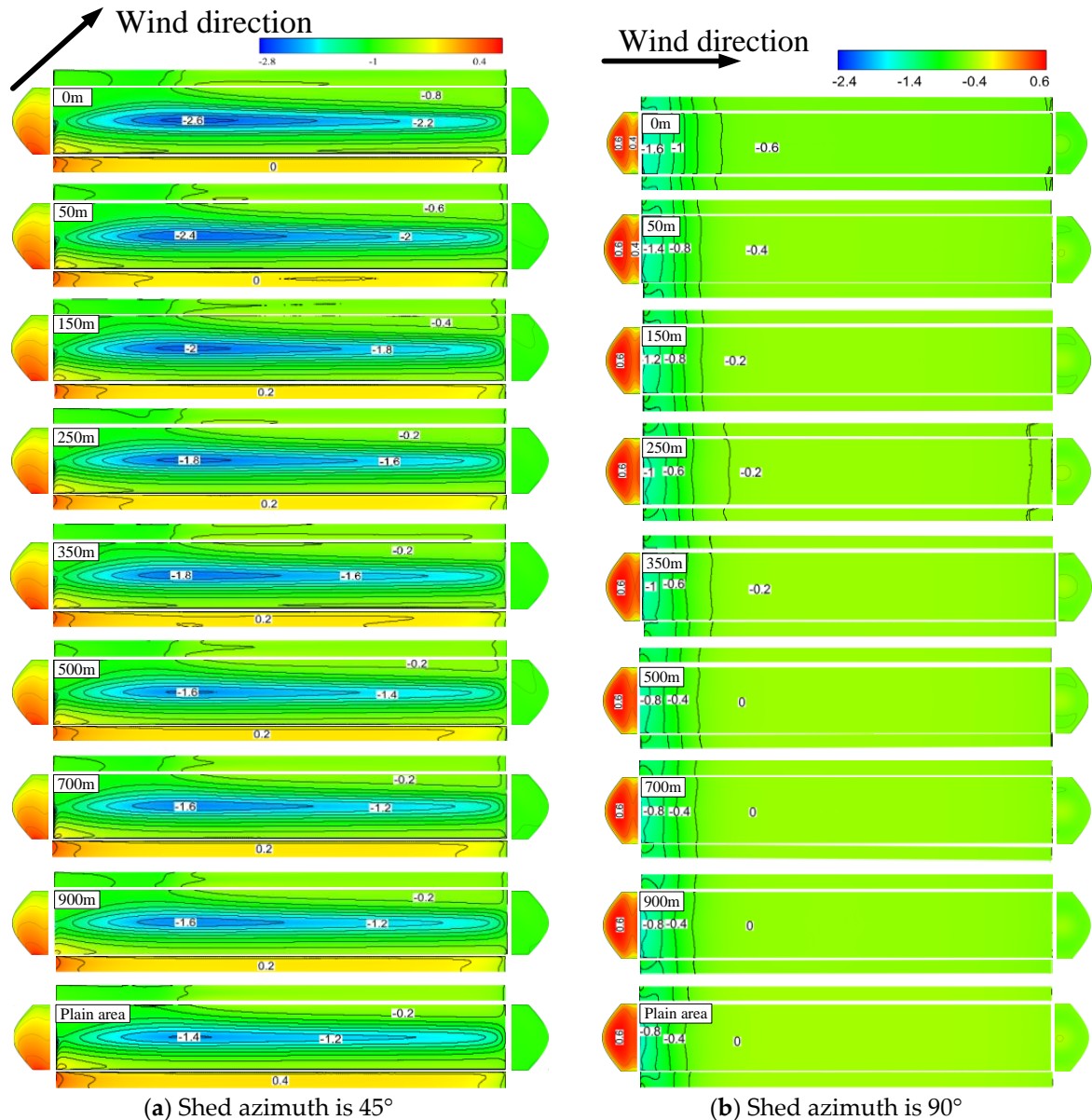

(**a**) Shed azimuth is 45°　　　(**b**) Shed azimuth is 90°

**Figure 13.** Wind pressure distribution of shed with *d* = 0, 50, 150, 250, 350, 500, 700, and 900 m and in a plain area.

**Table 6.** Distribution coefficient of wind pressure on each shed surface when the shed azimuth angle in the plain area is 45°.

| Area | MF | LF | RF | LW1 | LW2 | LW3 | RW1 | RW2 | RW3 | LB | MB | RB |
|------|------|------|------|------|------|------|------|------|------|------|------|------|
| Value | 0.272 | 0.410 | 0.186 | 0.651 | 0.358 | 0.054 | −0.333 | −0.342 | −0.358 | −0.348 | −0.119 | −0.076 |
| **Area** | **LT1** | **MT1** | **RT1** | **LT2** | **RT2** | **MT2** | **LT3** | **MT3** | **RT3** | | | |
| Value | −0.043 | −0.210 | −0.178 | −0.71 | −0.744 | −0.961 | −0.444 | −0.261 | −0.143 | | | |

**Table 7.** Distribution coefficient of wind pressure on each surface when the shed azimuth angle in the plain area is 90°.

| Area | LF/LB | MF/MB | RF/RB | LW1/LW3 | LW2 | RW1/RW3 | RW2 | LT1/LT2/LT3 | MT1/MT2/MT3 | RT1/RT2/RT3 |
|------|-------|-------|-------|---------|-----|---------|-----|-------------|-------------|-------------|
| Value | −0.388 | −0.017 | −0.037 | 0.405 | 0.668 | −0.120 | −0.086 | −0.405 | −0.019 | −0.035 |

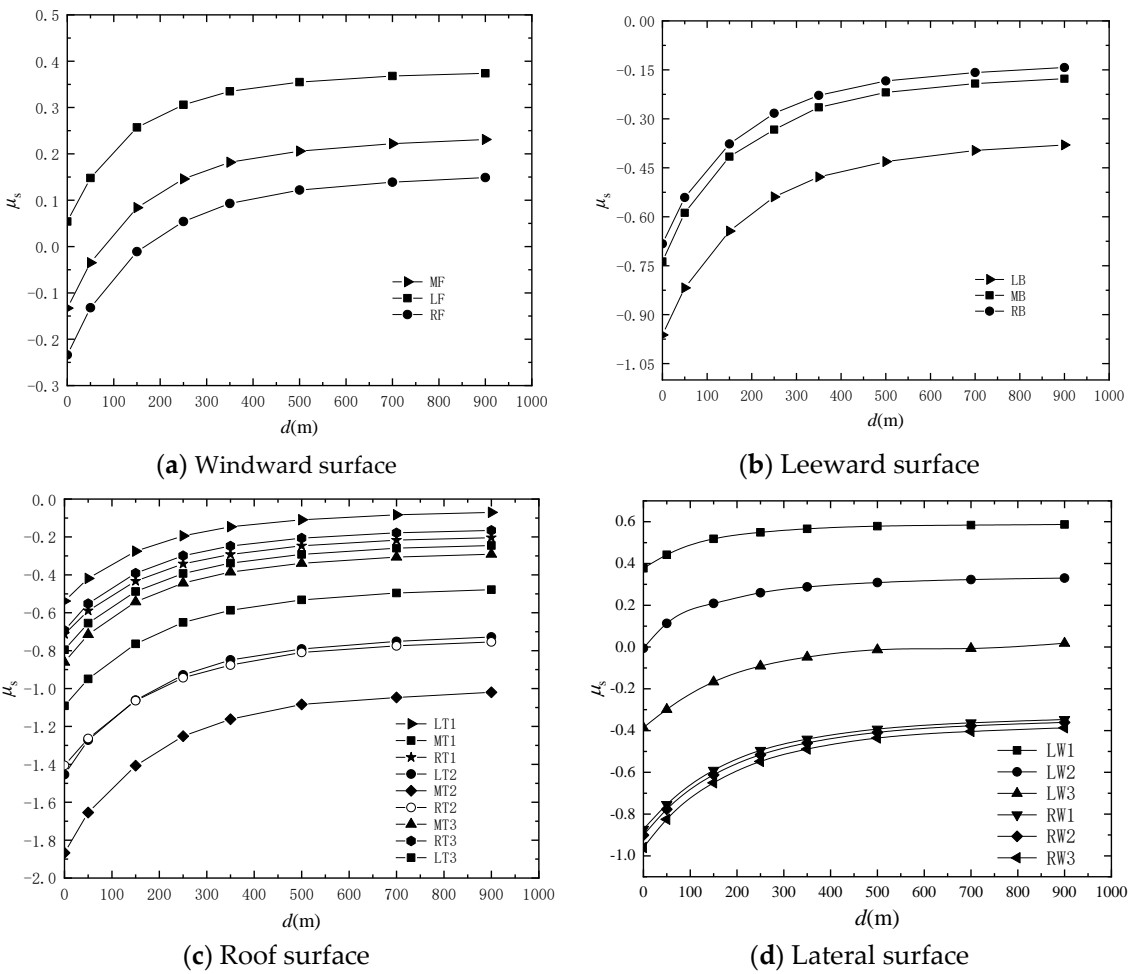

**Figure 14.** Wind pressure coefficient in each area of the greenhouse with 45° azimuth.

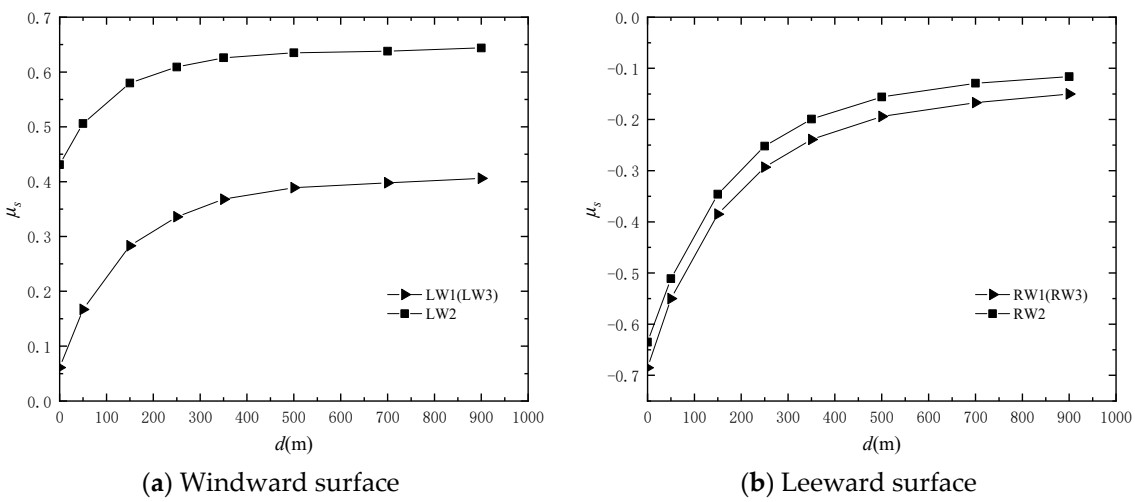

**Figure 15.** *Cont.*

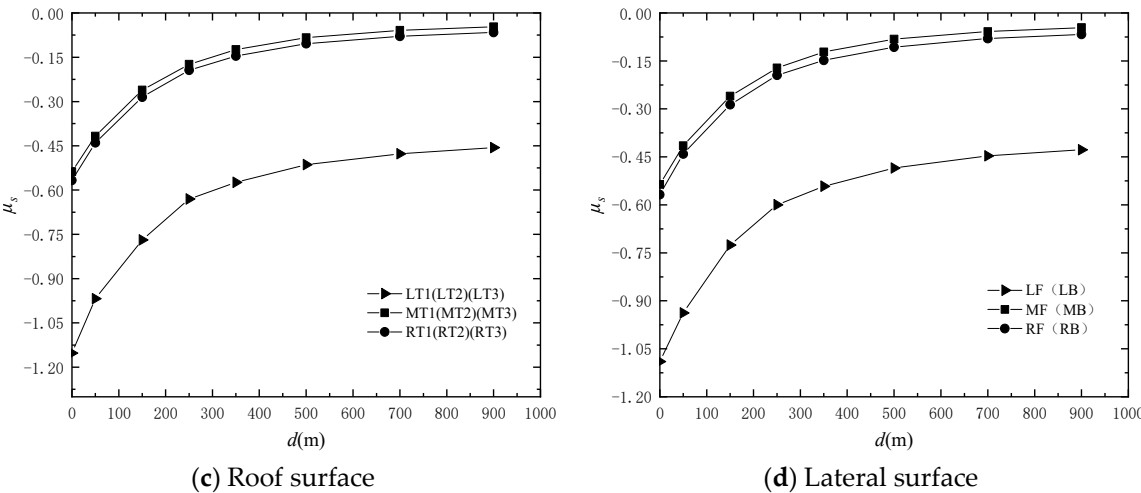

**Figure 15.** Wind pressure coefficient in each area of the greenhouse with 90° azimuth.

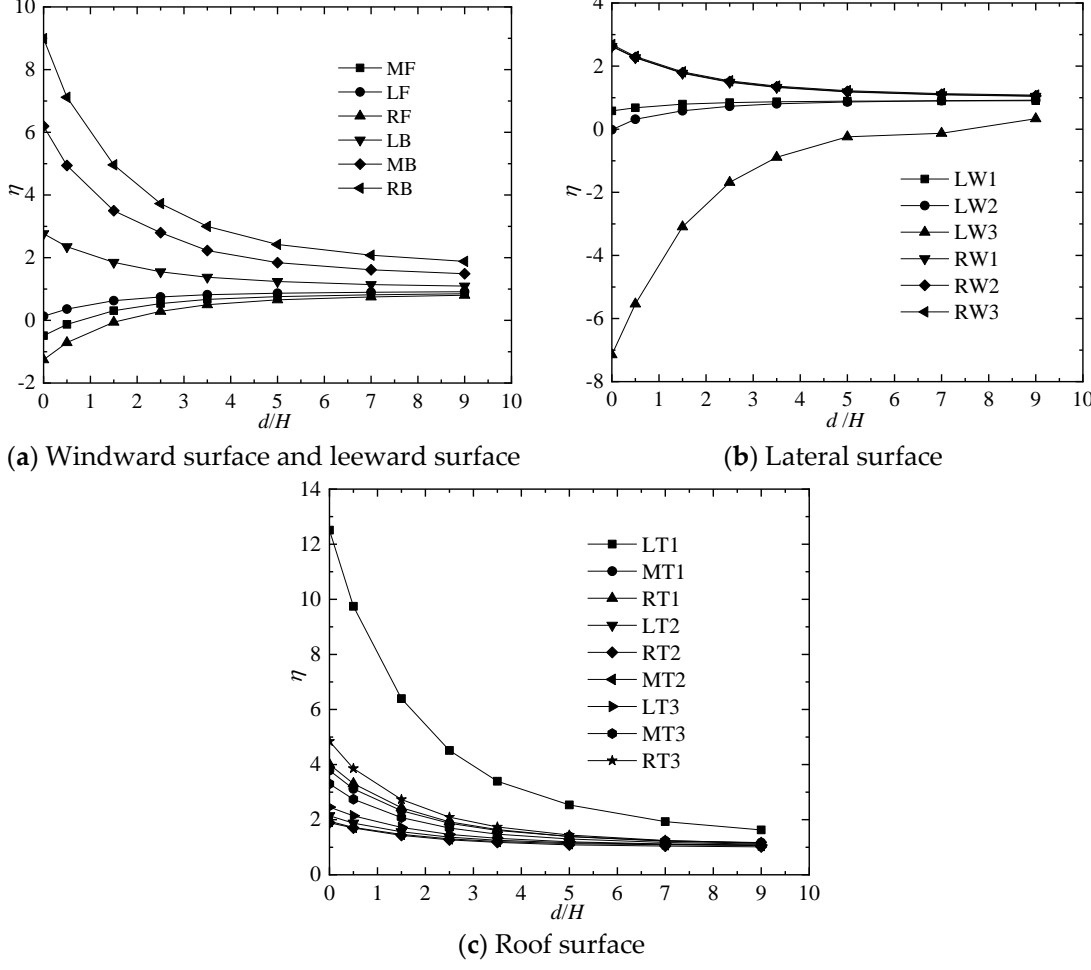

**Figure 16.** Relative value of wind pressure coefficient in valleys to that in plain areas considering 45° greenhouse azimuth.

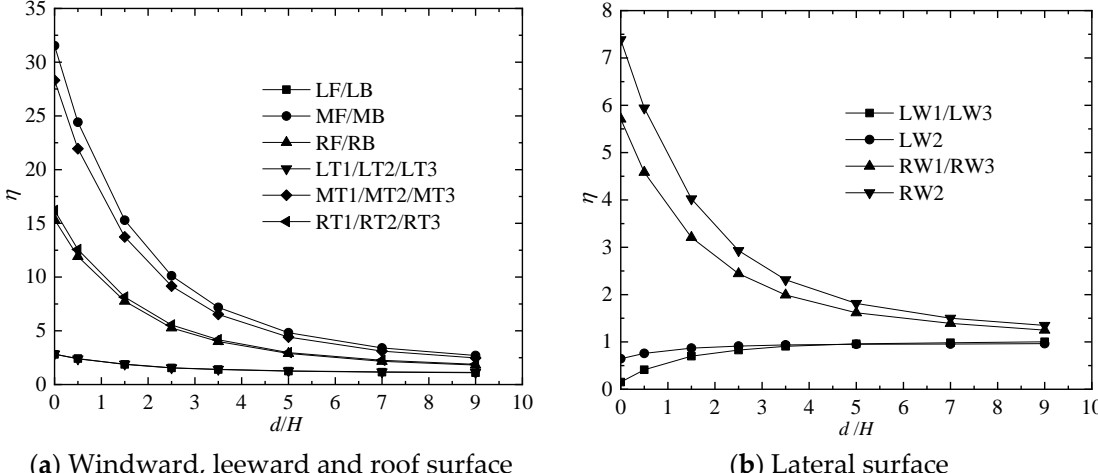

**(a)** Windward, leeward and roof surface    **(b)** Lateral surface

**Figure 17.** Relative value of wind pressure coefficient in valleys to that in plains area considering 90° greenhouse azimuth.

### 4.2. Analysis of Results from Computational Fluid Dynamics

As seen in Figure 13a, the greatest positive pressure is on the windward and lateral sides, considering the 45° shed azimuth. The greatest negative pressure is on the edge of the roof, and a larger negative pressure will also be generated in the LT2/MT2/RT2 area. As seen in Figure 13b, the greatest positive pressure is on the windward side, considering the 90° shed azimuth. The greatest negative pressure is generated in the LT1/LT2/LT3/LF/LB area.

Figure 13 also indicates that as the distance $d$ increases, the positive pressure on the windward side increases and the negative pressure on the side walls, roof surface, and leeward side decreases. When $d$ is less than 500 m, the wind pressure on all surfaces of the greenhouse shows obvious changes. When $d$ is larger than 500 m, the changes in wind pressure coefficient tend to be flat. When $d$ is within a certain range of values, the wind pressure coefficients on the greenhouse approach those in a plain area, as seen in Figure 13.

For the windward side, when $d$ is less than 50 m, the area LF is affected by the positive wind pressure, and the area MF and RF are affected by the negative wind pressure (Figure 14a). When $d$ is between 50 and 150 m, the wind pressure coefficient is positive on the LF/MF area, while the wind pressure coefficient is negative on the RF area. When $d$ is greater than 150 m, positive wind pressure appears on the windward side, and the wind pressure coefficient in area LF is greater than that in area MF and RF. The minimum wind pressure value appears on area RF. It is also shown that as distance $d$ increases, the wind pressure on the windward surface changes from negative to positive and is close to that in a plain region. A negative wind load appears on the leeward side (Figure 14b), and the wind load is slightly larger in the area LB than in the areas MB and RB. The roof surface of the greenhouse is affected by the negative wind pressure (Figure 14c). The highest suction load occurs in area MT2, followed by the area LT2/RT2, and the lowest wind pressure occurs in area LT1. The area MT1/RT1/MT3/RT3 bears the same wind load. The distance $d$ has a great influence on the wind load of the roof surface. Figure 14d shows that the windward crosswind areas LW1/LW2 are mainly affected by the positive wind pressure (except when $d$ is 0 m). The area LW3 is mainly affected by negative wind pressure. The positive pressure on area LW1 is the largest, followed by that on area LW2.

Windward side areas of LW1/LW2/LW3 are all subject to positive wind pressure (Figure 15a), and the wind pressure coefficients on the area LW1 are greater than those on the area LW2/LW3. With the increase in distance $d$, the wind pressure of the windward side is close to that in the plain area. Negative wind pressure appears on the leeward side, and the same value of pressure appears in areas RW1/RW3 and RW2 (Figure 15b). The wind pressure on the leeward side of the greenhouse is greatly affected by the distance $d$.

Negative wind pressure occurs on the roof surface (Figure 15c). The largest wind pressure appears on area LT1/LT2/LT3, followed by area RT1/RT2/RT3/MT1/MT2/MT3, which have roughly the same pressure values. Negative wind pressures appear in areas LF/LB, MF/MB, and RF/RB (Figure 15d). The analysis shows that the wind impact on the lateral side is similar to that on the roof surface of the greenhouse.

Figures 16 and 17 show that the ratio $d/H$ has a great impact on the wind pressure coefficient. If the ratio $d/H$ is less than 5, the ratio of coefficients derived from the valley area and the plain area is very large; when $d/H$ is larger than 5, the ratio values maintain stable; when $d/H$ is equal to 9, the ratio of wind pressure coefficients is approximately equal to 1, which means the wind pressure coefficients in the valley are close to those in the plain region.

### 4.3. Calculation Model

As can be seen from Figures 14–17, the wind pressure coefficients need to be calculated according to the figures. To clearly show the relationship between the wind pressure coefficient $\mu_s$ in valley areas and the distance $d$, the calculation model is proposed according to Figures 14 and 15, which can be expressed as:

$$\mu_s = A - Be^{-d/C} \tag{2}$$

where $A$, $B$, and $C$ are the fitting coefficients for different surfaces of a greenhouse, shown in Tables 8 and 9. As they are huge, only base data from the simulation and proposed calculation model considering 90° shed azimuth are shown in Table 10. It is shown that most of the relative errors between the calculation results from the proposed model and the simulation model (Figures 14 and 15) are within 15%. Some of the simulation results are too small and close to zero, resulting in high relative error, but these results have no impact on the design and safety of the greenhouse, and these errors can be ignored. Table 10 indicates the accuracy and reliability of the calculation model.

The formulas can be easily used to calculate the wind pressure coefficient with different mountain spacing and can serve as a supplement for the revision of specifications. In further research, the relationship between wind pressure and the ratio $d/H$ will also be discussed to provide a more comprehensive analysis.

**Table 8.** The values of the fitting coefficients when the azimuth angle of the shed is 45°.

| Coefficient | MF/RF | LF | LW1 | LW2 | LW3 | RW1/RW2/RW3/LB |
|---|---|---|---|---|---|---|
| *A* | 0.187 | 0.370 | 0.585 | 0.323 | 0.014 | −0.368 |
| *B* | −0.368 | −0.313 | −0.207 | −0.319 | −0.402 | −0.552 |
| *C* | 170.882 | 152.106 | 137.584 | 143.839 | 188.588 | 198.077 |

| Coefficient | MB/RB | LT2/RT2 | MT2 | LT3 | LT1/RT1/MT1/MT3/RT3 | |
|---|---|---|---|---|---|---|
| *A* | −0.136 | −0.865 | −0.750 | −0.293 | −0.233 | |
| *B* | −0.368 | −0.778 | −0.654 | −0.561 | −0.527 | |
| *C* | 170.883 | 193.447 | 206.595 | 187.287 | 189.639 | |

**Table 9.** The values of the fitting coefficients when the azimuth angle of the shed is 90°.

| Coefficient | LF/LB | MF/MB/RF/RB/MT1/MT2/MT3/RT1/RT2/RT3 | LW1/LW3 | LW2 | RW1/RW2/RW3 | LT1/LT2/LT3 |
|---|---|---|---|---|---|---|
| *A* | −0.4298 | −0.058 | 0.401 | 0.639 | −0.135 | −0.460 |
| *B* | −0.65971 | −0.491 | −0.337 | −0.206 | −0.522 | −0.683 |
| *C* | 190.05243 | 183.987 | 146.113 | 122.009 | 187.005 | 185.186 |

**Table 10.** The relative error between the results from the simulation and the calculation model considering 90° azimuth.

| d (Uint: m) | LW1/LW3 | | | LW2 | | | LF/LB | | |
|---|---|---|---|---|---|---|---|---|---|
| | SR | CMR | RE | SR | CMR | RE | SR | CMR | RE |
| 0 | 0.061 | 0.064 | 4.918% | 0.431 | 0.433 | 0.464% | −1.09 | −1.090 | 0.000% |
| 50 | 0.167 | 0.162 | 3.197% | 0.506 | 0.502 | 0.739% | −0.938 | −0.937 | 0.072% |
| 150 | 0.283 | 0.280 | 0.961% | 0.58 | 0.579 | 0.215% | −0.726 | −0.730 | 0.518% |
| 250 | 0.336 | 0.340 | 1.223% | 0.609 | 0.612 | 0.567% | −0.6 | −0.607 | 1.186% |
| 350 | 0.368 | 0.370 | 0.622% | 0.626 | 0.627 | 0.208% | −0.542 | −0.535 | 1.356% |
| 500 | 0.389 | 0.390 | 0.257% | 0.635 | 0.636 | 0.091% | −0.485 | −0.478 | 1.540% |
| 700 | 0.398 | 0.398 | 0.050% | 0.638 | 0.638 | 0.053% | −0.447 | −0.447 | 0.091% |
| 900 | 0.406 | 0.400 | 1.407% | 0.644 | 0.639 | 0.796% | −0.428 | −0.436 | 1.821% |
| 10,000 | 0.405 | 0.401 | 0.988% | 0.668 | 0.639 | 4.341% | −0.388 | −0.430 | 10.825% |

| d (Uint: m) | MF/MB | | | RF/RB | | | RW1/RW3 | | |
|---|---|---|---|---|---|---|---|---|---|
| | SR | CMR | RE | SR | CMR | RE | SR | CMR | RE |
| 0 | −0.536 | −0.549 | 2.425% | −0.568 | −0.549 | 3.345% | −0.685 | −0.657 | 4.088% |
| 50 | −0.415 | −0.432 | 4.136% | −0.441 | −0.432 | 2.004% | −0.55 | −0.535 | 2.812% |
| 150 | −0.26 | −0.275 | 5.877% | −0.287 | −0.275 | 4.084% | −0.385 | −0.369 | 4.142% |
| 250 | −0.172 | −0.184 | 7.079% | −0.195 | −0.184 | 5.551% | −0.293 | −0.272 | 7.129% |
| 350 | −0.122 | −0.131 | 7.599% | −0.148 | −0.131 | 11.303% | −0.239 | −0.215 | 9.907% |
| 500 | −0.082 | −0.090 | 10.273% | −0.107 | −0.090 | 15.491% | −0.194 | −0.171 | 11.848% |
| 700 | −0.058 | −0.069 | 18.852% | −0.08 | −0.069 | 13.832% | −0.167 | −0.147 | 11.761% |
| 900 | −0.046 | −0.062 | 34.103% | −0.067 | −0.062 | 7.930% | −0.15 | −0.139 | 7.172% |
| 10,000 | −0.017 | −0.058 | 241.176% | −0.037 | −0.058 | 56.757% | −0.12 | −0.135 | 12.500% |

| d (Uint: m) | RW2 | | | LT1/LT2/LT3 | | | MT1/MT2/MT3 | | |
|---|---|---|---|---|---|---|---|---|---|
| | SR | CMR | RE | SR | CMR | RE | SR | CMR | RE |
| 0 | −0.536 | −0.549 | 2.425% | −0.568 | −0.549 | 3.345% | −0.685 | −0.657 | 4.088% |
| 50 | −0.415 | −0.432 | 4.136% | −0.441 | −0.432 | 2.004% | −0.55 | −0.535 | 2.812% |
| 150 | −0.26 | −0.275 | 5.877% | −0.287 | −0.275 | 4.084% | −0.385 | −0.369 | 4.142% |
| 250 | −0.172 | −0.184 | 7.079% | −0.195 | −0.184 | 5.551% | −0.293 | −0.272 | 7.129% |
| 350 | −0.122 | −0.131 | 7.599% | −0.148 | −0.131 | 11.303% | −0.239 | −0.215 | 9.907% |
| 500 | −0.082 | −0.090 | 10.273% | −0.107 | −0.090 | 15.491% | −0.194 | −0.171 | 11.848% |
| 700 | −0.058 | −0.069 | 18.852% | −0.08 | −0.069 | 13.832% | −0.167 | −0.147 | 11.761% |
| 900 | −0.046 | −0.062 | 34.103% | −0.067 | −0.062 | 7.930% | −0.15 | −0.139 | 7.172% |
| 10,000 | −0.017 | −0.058 | 241.176% | −0.037 | −0.058 | 56.757% | −0.12 | −0.135 | 12.500% |

| d (Uint: m) | RT1/RT2/RT3 | | |
|---|---|---|---|
| | SR | CMR | RE |
| 0 | −0.567 | −0.549 | 3.175% |
| 50 | −0.44 | −0.432 | 1.781% |
| 150 | −0.285 | −0.275 | 3.411% |
| 250 | −0.194 | −0.184 | 5.064% |
| 350 | −0.146 | −0.131 | 10.088% |
| 500 | −0.104 | −0.090 | 13.054% |
| 700 | −0.079 | −0.069 | 12.742% |
| 900 | −0.066 | −0.062 | 6.535% |
| 10,000 | −0.035 | −0.058 | 65.714% |

Note: SR and CMR represent the simulation result and calculation model result, respectively; RE represents relative error.

## 5. Conclusions

In this study, the wind pressure of a single-span arched plastic greenhouse with different shed azimuths is generated based on the validated CFD model. The conclusions are as follows: when the azimuth angle of the greenhouse is 45°, the greatest positive wind pressure is generated at the windward side of the side walls, and the greatest suction power is located at the edge of the shoulder of the windward side of the roof surface of

the greenhouse. When the azimuth angle of the greenhouse is 90°, the greatest positive pressure is generated on the windward side, and the greatest suction load is located at the junction edge of the windward and the roof surface. The positive wind pressure power increases, and the suction power decreases with an increase in $d$. When the $d/H$ ratio is less than five, the wind pressure coefficient on each surface of the greenhouse changes, and the changes then plateau when the ratio is greater than five. After the $d/H$ ratio reaches nine, the wind pressure on the greenhouse located in a valley tends to be close to that in the plain area.

The proposed calculation models fit well with the variation rule of wind pressure coefficients in different areas of a plastic greenhouse, which show that the calculation model can be effectively used to calculate the wind pressure coefficients of the single-span arched greenhouse. This research will be valuable for the design and optimization of single-span arched greenhouses in valley areas. The research results and the proposed calculation model may also be helpful for the revision or supplementation of China's load codes.

**Author Contributions:** Conceptualization, Z.L. and J.X.; software, G.H.; investigation, Z.L., Y.L., Z.G. and Q.W.; data curation, Z.L. and J.X.; writing—original draft preparation, X.R.; writing—review and editing, J.X.; visualization, G.H. and J.X.; supervision, S.Z. All authors have read and agreed to the published version of the manuscript.

**Funding:** This research work was funded by National Natural Science Foundation of China (U20A2020) and Beijing Municipal Natural Science Foundation (8214051).

**Institutional Review Board Statement:** No applicable.

**Informed Consent Statement:** No applicable.

**Data Availability Statement:** No applicable.

**Conflicts of Interest:** The authors declare no conflict of interest.

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
