# Peer review of "Analysis of Wind Pressure Coefficients for Single-Span Arched Plastic Greenhouses Located in a Valley Region Using CFD"

_agronomy, doi:10.3390/agronomy13020553_

Round 1

Reviewer 1 Report

The authors have studied the wind pressure of a greenhouse in a valley. The mountain terrain is modeled with simplified sine form, and wind pressure is evaluated using CFD modeling. The work is well structured, and I would like the authors to take care of the following aspects before the manuscript can be accepted.

1. The authors have missed a lot of recent work on wind pressure measurements on greenhouse roofs like some of the listed below

Maraveas, C. (2020). Wind pressure coefficients on greenhouse structures. Agriculture10(5), 149.

Bournet, P. E., & Rojano, F. (2022). Advances of Computational Fluid Dynamics (CFD) applications in agricultural building modelling: Research, applications and challenges. Computers and Electronics in Agriculture201, 107277.

Wang, C., Nan, B., Wang, T., Bai, Y., & Li, Y. (2021). Wind pressure acting on greenhouses: A review. International Journal of Agricultural and Biological Engineering14(2), 1-8.

2. Better to represent the fitted equation with base data with error limits.

3. As looked at the literature in the said area, most of the time, the researchers look at the wind pressure. But one aspect that they have neglected, the nature of wind pressure obtained on the roof of the greenhouse, is due to flow separation. It would be nice of the authors to look into this aspect when they are explaining the results.

Reviewer 2 Report

Similarity in this paper is higher than the standard of the Journal. A single source similarity is 23% which is not acceptable at all. Authors need to rewrite the complete article and explain the novelty and research contribution in this article. In addition, author should also carefully look into the following suggestions.

1.      Introduction section is very weak and the authors missed the latest relevant research. Most of the literature is outdated or only cited without explanation. Author should discuss the finding of the earlier research and also discuss the limitations of those studies. The literature study can be enhanced by adding recent relevant references. I would say that Introduction section is the weakest part in this article.

2.      Check the manuscript for grammatical errors

3.      Article is written in non-technical way. For example, the sentences like model is made in SPACE Claim; model is imported into the Fluent Meshing;..No need to mention ANSYS module.

4.      Very less information is provided related to mesh study. Information related to y+ value is mandatory in CFD studies. Authors should go in detail for relation between y+ and turbulent models.

Reviewer 3 Report

Dear Authors,

The evaluated manuscript is focused on the analysis of the effect of wind speed in built greenhouses. The main objective was to promote the understanding of the characteristics of the wind and guarantee the structural safety of the greenhouse in the valley area and for this it was considered pertinent to analyze the distribution law of the wind pressure in the greenhouse. As the main result of this reserach, relationship curves between the distance from the mountain and the coefficient of wind pressure are proposed. This work succeeded in showing for the studied region that there is a canyon wind effect in the valley area, and the influence of canyon wind on the change of wind pressure in each plastic greenhouse area should be considered in the structural design.

After carefully reading your manuscript, I have some suggestions that are listed in the following lines:

1.  Please review equation 3 because it is not correct to write as an equality between equations for a continuous variable with the case for a discrete variable.

2. It is also important to facilitate the reader's understanding the inclusion of a section explaining the system of equations that must be solved by CFD and justifying its applicability. Also, could you explain in sufficient detail why you selected the realizable K-epsilon as your turbulence model?

3. In figure 8 could you include a third image of the differences between figures a and b? and use statistics to explain these differences.

4. Try to use the same range of scales, where applicable, to graphically compare the differences between these figures. (i.e. case of figures 9 and 12)

Best regards

Round 2

Reviewer 2 Report

1. Revised version still has higher similarity ratio from a single source (more than 5%).
2. This is a simple case study. How author justify the novelty and research gap?
3. Introduction section is not improved. Only citing the references is not sufficient. Need to explain the latest research carried out in the last few years.
4. Most of the comments which were raised previously, are not addressed properly.

Round 3

Reviewer 2 Report

Reviewer comments have been addressed and the article is suitable for publication.